# Overexpression of a *Fragaria* × *ananassa* AP2/ERF Transcription Factor Gene (*FaTINY2*) Increases Cold and Salt Tolerance in *Arabidopsis thaliana*

**DOI:** 10.3390/ijms26052109

**Published:** 2025-02-27

**Authors:** Wenhui Li, Wenhao Zhang, Huiwen Li, Anqi Yao, Zhongyong Ma, Rui Kang, Yanbo Guo, Xingguo Li, Wenquan Yu, Deguo Han

**Affiliations:** 1Key Laboratory of Biology and Genetic Improvement of Horticultural Crops (Northeast Region), Ministry of Agriculture and Rural Affairs, National-Local Joint Engineering Research Center for Development and Utilization of Small Fruits in Cold Regions, College of Horticulture & Landscape Architecture, Northeast Agricultural University, Harbin 150030, China; wenhuili@neau.edu.cn (W.L.); 18346971459@163.com (W.Z.); lhw989874@163.com (H.L.); ann16653765531@163.com (A.Y.); 18132797609@163.com (Z.M.); m18298834887@163.com (R.K.); 15655436568@163.com (Y.G.); xingguoli@neau.edu.cn (X.L.); 2Mudanjiang Branch of Heilongjiang Academy of Agricultural Sciences, Mudanjiang 157041, China

**Keywords:** *FaTINY2*, AP2/ERF, low temperature, high salt

## Abstract

The AP2/ERF family of transcription factors is one of the most conserved and important transcription factor families, and it is ubiquitous in plants. It plays an essential role in plant morphogenesis, molecular mechanisms of stress responses, hormone signaling pathways, and synthesis of secondary metabolites. *FaTINY2* was cloned from the octaploid strawberry *Fragaria* × *ananassa* for this investigation. Bioinformatics revealed that the protein possesses a conserved AP2 domain and is localized in the nucleus. When *FaTINY2* was expressed in plants, quantitative analysis revealed that the gene was tissue-specific. There are lower contents of reactive oxygen species (ROS) and malondialdehyde (MDA), higher contents of proline, chlorophyll, and higher activities of catalase (CAT), superoxide dismutase (SOD), and peroxidase (POD) in transgenic *Arabidopsis thaliana* than wild type (WT) and unload line (UL) plants under cold and salt stress. FaTINY2 plays a role in enhancing stress tolerance by regulating a few genes linked to the stress response. The findings of this study were that *FaTINY2* transgenic *Arabidopsis thaliana* plants were more tolerant to salt and cold than WT and UL plants. In addition to offering a theoretical reference for strawberry production under stress, this research established a groundwork for exploration into the molecular mechanisms in which strawberries respond to cold and high salt stress.

## 1. Introduction

Plant growth and development can be affected significantly due to abiotic and biological stressors [1,2]. Abiotic stressors like low temperature (LT), high temperature, high salt, nutrient deficiency, heavy metal excess, ozone, and UV radiation will have adverse effects on plant morphogenesis and result in a significant decrease in yield [3,4,5]. In agricultural production, low temperature and high salt are the primary abiotic stressors, which not only influence the ecological distribution and growth patterns of plants but also impact various physiological processes and biochemical reactions within plant cells [6,7]. Low-temperature stress is typically categorized into two forms: cold damage and freezing damage [8]. Exposure of plants to low temperatures leads to notable changes in their morphology, characterized by a reduced growth rate, such as the extension of leaves and the elongation of stems, which are inhibited. This condition also results in leaf wilting. Additionally, there are alterations in membrane permeability, resulting in increased electrolyte exosmosis [9]. Reduced temperatures lead to a rise in the levels of reactive oxygen species (ROS), and the accumulation of ROS will lead to an increase in the expression level of membrane lipid peroxide MDA, thus affecting the normal function of cells [10,11]. In high-salt environments characterized by elevated osmotic potential due to chloride ions (Cl^−^) and sodium ions (Na^+^), plant physiology is significantly impacted [12]. High salt stress disrupts photosynthesis, halts chloroplast function, accelerates degradation processes, causes leaf discoloration, and restrains overall plant growth [13]. This condition leads to water absorption difficulties, produces ionic toxic effects, disruption of ion homeostasis, metabolic disorders, and an increase in reactive oxygen species (ROS) levels. Additionally, it raises the osmotic pressure of soil solutions, weakens the water absorption capacity in the root system, and contributes to physiological drought conditions [14,15].

The AP2/ERF (APETALA2/ethylene-responsive element binding factors) family is one of the most conserved and important transcription factor families, and it is ubiquitous in plants [16]. The members of this superfamily are crucial in diverse biological functions, including the development of plants, stress response mechanisms, hormone signaling, and metabolic regulation [17,18,19,20,21]. Generally, transcription factors of AP2/ERF contain an AP2 domain, which comprises 60 to 70 amino acids [22,23,24]. Considering the quantity of AP2 domains and sequence homology, this family is subdivided into three primary subfamilies, namely AP2, RAV, ERF, and Soloist (with a few unclassified factors) [25,26,27,28]. Among them is the distinctiveness of the AP2 family in its dual AP2 domains. In contrast, the ERF family features a single AP2 domain, while the RAV family is distinguished by having both one AP2 and one B3 domain [29,30,31]. The division of the ERF family into the ERF and DREB (dehydration reaction element binding) subfamilies depends on the differences between the 14th and 19th amino acids of the AP2 domain, categorized into B1–B6 and A1–A6 groups, respectively [32,33,34]. The main function of DREB proteins is to bind an A/GCCGAC element in response to abiotic stress, while ERF proteins are crucial in this process through their specific binding to AGCCGCC elements (GCC box) [35].

DREB1 transcription factor is an A1 group transcription factor, most of which are transcription factors responding to cold injury and dehydration [36]. The A2 group is mainly the DREB2 transcription factor, which responds to high salt and drought stress but is not induced by low-temperature stress [37,38]. The A3 group is mainly represented by ABI4 (ABSCISIC ACID INSENSITIVE 4), which is involved in ABA signaling and glucose signaling [39]. At present, only two genes, *TINY* and *TINY2*, have been reported in the *Arabidopsis thaliana* (*A. thaliana*) A4 group. Among them, the expression of *TINY* is largely affected by drought, cold damage, and ethylene activation [40]. In addition, *TINY* is also activated by methyl jasmonate and can negatively regulate plant growth by inhibiting the BR pathway [41]. *TINY2* in *A. thaliana* is induced by ABA, cold, drought, and high salt [42,43]. However, members of the A5 and A6 groups only remained in the expression level analysis of a few genes.

Numerous AP2/ERF transcription factors (TFs) have been detected in strawberries [44]. Moreover, these TFs participate in the growth and development process of strawberries and play a key role in the response of strawberries to biological and abiotic stresses [45]. This research identified a member of the DREB subfamily that is part of the AP2/ERF superfamily, which was most similar to *AtTINY2* in *A. thaliana*, so the transcription factor gene was named *FaTINY2*, and the involvement in the stress response in *A. thaliana* was verified. The investigation revealed that low temperature and high salt were the two main abiotic stress factors inducing *FaTINY2* expression. *FaTINY2* transgenic *A. thaliana* was used as experimental material to substantiate that the upregulated expression of *FaTINY2* significantly boosts the cold resistance and salt tolerance of transgenic *A. thaliana.* This finding establishes a foundational theoretical groundwork for subsequent explorations into the molecular regulatory mechanisms underlying the influence on plant stress resilience of *FaTINY2*.

## 2. Results

### 2.1. Cloning and Bioinformatic Analysis of FaTINY2

In this study, *FaTINY2* (*FaERF25*, MH332927.1), an AP2/ERF TF gene, was isolated from *Fragaria* × *ananassa* with a total length of 762 bp. Bioinformatics studies suggested that the FaTINY2 protein encodes 253 amino acids, with a theoretical molecular weight estimated at 27,487.54 Da. In the amino acid composition, Ser (18.6%), Leu (9.1%), Ala (7.1%), and Pro (7.1%) had higher proportions. In total, there are 31 negatively charged residues (Asp + Glu) and 25 positively charged residues (Arg + Lys). Since the grand average of hydropathicity of FaTINY2 is −0.495, it is classified as a hydrophilic protein.

The focus of this study was to compare the amino acid sequences of FaTINY2 protein with TINY proteins in other species. Through this comparative approach, a phylogenetic tree has been constructed to clarify the evolutionary relationships and homologies among these proteins. FaTINY2 has the highest homology similarity with RrTINY (*Rosa rugosa*, XP_062025076.1) and belongs to the same evolutionary branch in the phylogenetic tree (Figure 1A,B). SOPMA is used to predict the secondary structure of proteins, and predicted the secondary structure of FaTINY2, showing that the secondary structure of FaTINY2 consisted of 21.74% α-helix, 1.58% β turn, 70.75% random coil 5.93%, and extended strands (Figure 1C,D).

### 2.2. FaTINY2 Was Localizated onto Nucleus

In order to obtain subcellular localization of FaTINY2, we constructed 35S::FaTINY2-GFP plasmid and transfected *Agrobacterium*. *Agrobacterium* was then injected into the tobacco leaf outer epidermal cells (Figure 2). The fluorescence of leaf cells was observed using laser confocal microscopy. Findings indicated that the fluorescence of the 35S::FaTINY2-GFP fusion protein was exclusive to the nucleus (Figure 2E), with fluorescence observable in both cell membrane and nucleus of control tobacco infected with empty plasmid (35S::GFP) (Figure 2A). The further DAPI staining confirmed the nuclear localization of the FaTINY2 protein.

### 2.3. Expression Analysis of FaTINY2

The expression level of *FaTINY2* in various strawberry tissues, such as young leaves, mature leaves, stems, and roots, was measured using Real-Time quantitative PCR (RT-qPCR). Findings indicated that the presence of *FaTINY2* expression in the expression level of young leaves and roots was significantly higher than that found in stems and mature leaves (Figure 3A). *FaTINY2* shows a higher sensitivity to LT, salt, and dehydration stress. The treatment of strawberry seedlings involved LT, high salt, dehydration, high temperature, and ABA stresses, respectively, there was an initial rise and subsequent decline in *FaTINY2* expression in the young leaves and roots. In young leaves, the maximum expression of *FaTINY2* was 2 h, 8 h, 8 h, 2 h, and 6 h under each abiotic stress treatment. In roots, *FaTINY2* expression reached its maximum at 4 h, 2 h, 4 h, 6 h, and 2 h. When subjected to LT, high salt, and dehydration stress, young leaves and roots exhibited greater *FaTINY2* expression compared to other treatments (Figure 3B,C). In conclusion, cold stress, salt stress, and dehydration stress have significant induction effects on *FaTINY2*.

### 2.4. Overexpression of FaTINY2 Enhanced Cold Tolerance of Arabidopsis thaliana

Overexpression *FaTINY2* transgenic *A. thaliana* lines (L1, L2, L3, L4, L5) were constructed to explore the function of *FaTINY2* in cold tolerance and salt tolerance. FaTINY2 was expressed at a higher level in L1, L4, and L5 lines, so these three lines were selected for subsequent experimental treatment (Figure 4A). Without any treatment, the phenotypes of WT, UL, L1, L4, and L5 were basically the same (Figure 4B). At a temperature of −8 °C for a duration of 14 h, there was a notable alteration in the WT and UL line phenotypes, and the leaves were severely wrinkled. However, the development of the three overexpressed lines (L1, L4, L5) remained largely unaltered. Following a week-long recuperation at room temperature of 24 °C, the leaves of transgenic *A. thaliana* showed only slight shrinkage. Conversely, WT and UL suffered greater damage under cold stress, manifested as leaf wilting and even death (Figure 4B). After stress treatment, survival rates of WT and UL lines were 28% and 29%, respectively, while the survival rates of L1, L4, and L5 were 87%, 90%, and 89%, respectively (Figure 4C).

Figure 5 illustrates that at a control condition of 22 °C, the levels of chlorophyll, MDA, proline, H_2_O_2_, and O_2_^−^ contents, along with the activities of CAT, SOD, and POD were nearly identical, without significant differences. Following exposure to cold, the contents in both WT and UL lines, the levels of MDA, H_2_O_2_, and O_2_^−^ were significantly increased and significantly higher than those of *FaTINY2*-OE lines (L1, L4, and L5). And the contents of chlorophyll and proline, along with the activities of CAT, SOD, and POD, were found to be less than in *FaTINY2*-OE lines. The findings indicate that overexpression of *FaTINY2* may boost antioxidant enzyme activity and reduce membrane lipid peroxidation, thus improving cold tolerance of *A. thaliana*.

Investigating the molecular function of *FaTINY2* in plants under cold stress involves examining the expression levels of four specific genes associated with cold stress. *AtCBF1*, *AtCBF4*, *AtCOR15a*, and *AtCOR15b* were determined in this study. In contrast to WT and UL lines, *FaTINY2*-OE lines, when subjected to low temperature, exhibited significant up-regulation in cold-response gene expression (Figure 6), indicating that *FaTINY2* can enhance the cold tolerance of transgenic plants by regulating the expression of downstream stress-related genes.

### 2.5. Overexpression of FaTINY2 Enhanced Salt Tolerance of Arabidopsis thaliana

WT, UL, L1, L4, and L5 were irrigated with 200 mM NaCl solution for 7 days, succeeded by a three-day water treatment. Phenotypic observation showed that WT and UL plants were seriously affected under NaCl irrigation and showed a remarkable dehydration wilting phenomenon. The phenotypes of L1, L4, and L5 were not significantly changed, and the leaves were still green. Following the cessation of the 200 mM NaCl irrigation and three-day water treatment of the plants, the wilting of both WT and UL plants worsened, leading to death, and the survival rate of both lines was 33%. The three lines of *FaTINY2*-OE (L1, L4, and L5) showed only slight leaf shrinkage, and the damage was less severe than that of WT and UL plants, with survival rates of 88%, 88%, and 83%, respectively (Figure 7B,C). Findings indicated that the overexpression of *FaTINY2* could mitigate the harm inflicted by salt stress to some degree.

After NaCl treatment, the contents of physiological indicators associated with salt stress resistance are present, and these data show that WT and UL of the transgenic line were basically at the same level as those before stress treatment. Compared with WT and UL, the contents of chlorophyll, proline, and activities of antioxidant enzymes (CAT, SOD, POD) showed a significant increase in transgenic lines treated with 200 mM NaCl. Contents of MDA, H_2_O_2_, and O_2_^−^ in the transgenic lines were found to be less than those in WT and UL lines (Figure 8). Consequently, FaTINY2 can significantly increase the salt tolerance of transgenic plants. This research aimed to reveal the molecular regulation mechanism of salt stress through the measure of four genes associated with salt stress: *AtNHX1*, *AtSnRK2*, *AtKUP6*, and *AtKUP7* (Figure 9). The results revealed a significant increase in the expression levels of these four genes in L1, L4, and L5 transgenic lines in comparison to WT and UL lines under salt treatment. The findings imply that FaTINY2 could improve the tolerance of transgenic plants to high salt stress by regulating genes linked to stress.

## 3. Discussion

The research involved cloning *FaTINY2* from *Fragaria* × *ananassa*. Analysis of protein sequences revealed that FaTINY2 possesses a highly conserved AP2 domain. In sequence comparison, it was found that FaTINY2 had high homology similarity with RrTINY, MsTINY, and PbTINY. The reason could be that these species belong to the Rosaceae family [46]. So far, the subcellular localization of DREB TFs has been extensively studied. Findings from this research indicated FaTINY2 was localized in the nucleus, and that is consistent with the results of *OsDREBL* (*Oryza sativa* L.), *GmDREB1* (*Glycine max*), MdTINY (*Malus domestica*) [47,48,49].

Gene expression varies by tissue during the growth and development of plants, often manifesting as differences in gene expression levels among different tissues or organs [50]. This research examines the levels of *FaTINY2* expression across different organs in strawberry seedlings. *FaTINY2* is more expressed in young leaves and roots compared to stems and mature leaves, likely due to FaTINY2 being more sensitive to abiotic stress in young tissues, suggesting that the expression of *FaTINY2* is tissue-specific in different organs of *Fragaria* × *ananassa*. The tissue specificity found in plants such as *StDREB1* in potato (*Solanum tuberosum* L.), *PagERF021* in poplar (*Populus alba* × *P. glandulosa*), and *ZmDREB4.1* in maize (*Zea mays*) [51,52,53]. These are similar to the results of this study.

Plants cannot move, so they must endure cold, high salt, and other abiotic stressors [6]. These stress factors can increase the intracellular ROS content, potentially leading to significant harm to the cellular architecture. Key elements of the plant antioxidant mechanism, namely CAT, SOD, and POD, are capable of eliminating ROS from cells when stressed and lessening the damage ROS inflicts on cells [54]. Therefore, the activity of these enzymes can be used as an indicator of the extent to which plants are damaged in times of adversity. The content of chlorophyll, proline, and MDA frequently serve as indicators of resilience to stress in plants [55].

Sugar beet (*Beta vulgaris* L.) was treated at 4 °C for 12 h; there was a notable rise in MDA content and SOD activity, suggesting that LT can induce the expression of *BvDREB* [56]. Under the treatment of 100 mM NaCl solution, ROS accumulation in *ThDREB* transgenic lines was significantly reduced [57]. In conditions of drought, transgenic plants with increased *DREB46* expression exhibited enhanced resistance to stress, reduced water depletion, lower MDA levels, and higher chlorophyll content [58]. The common bean (*Phaseolus vulg*) was used to amplify a TINY-like AP2/ERF gene, *PvERF35*, which showed increased expression in tobacco and was treated with 200 mM NaCl. There was a notable rise in proline levels, resulting in enhanced tolerance to salt [59]. This research revealed that post-stress treatment, the chlorophyll and proline levels in transgenic *A. thaliana* exceeded those in WT and UL lines, suggesting reduced damage in the transgenic lines. Concurrently, the overexpression of *FaTINY2* amplified the functions of CAT, SOD, and POD when subjected to low temperature and high salt stress.

The ICE-CBF-COR route is vital for plants to endure and adapt to cold conditions [60]. Overexpression of *MbDREB1* leads to activation of *COR15a* expression, thereby enhancing the cold tolerance of transgenic *A. thaliana* [61]. In *Zoysia japonica* Steud, overexpression of *ZjDREB1.4* can activate the expression of downstream genes *COR15a*, *COR15b*, *RD29B*, and *COR413* [62]. Within *A. thaliana*, *AtCBF1* and *AtCBF4* bolster resilience against low-temperature stress through the activation of its subsequent genes *AtRD29A*, *AtKIN1*, *AtCOR15a*, and *AtCOR47* [63]. All these findings demonstrate the function of the CBF signaling pathway under cold stress. There is an increase in the expression levels of *ATCBF1*, *AtCBF4*, *AtCOR15a* [64], and *AtCOR15b* [65] under cold stress. It investigated the potential of *FaTINY2* to boost plant resistance to cold through the activation of various stress-associated genes.

When subjected to salt stress, *GmDREB2* gene expression induced downstream *Rd29A* and *COR15a* gene expression [66]. In transgenic salvia miltiorrhiza, the ectopic expression of *AtDREB1A* activates genes that respond to stress, like *KIN1* and *KIN2*, protective proteins like proline-rich protein 4, all of which intensify under stress [67]. In addition, the *NHX1* encoding Na^+^/H^+^ antiporter protein is involved in regulating the transport of Na^+^, thereby enhancing salt tolerance [68]. SnRK2, a crucial regulator of the ABA signaling pathway, has the ability to phosphorylate numerous subsequent target proteins post-activation, thus regulating various physiological processes in response to ABA [69,70,71]. KUP protein can indirectly affect ABA signaling pathways by regulating intracellular K^+^ homeostasis [72]. This study reveals that FaTINY2 may enhance salt tolerance in plants by activating a series of genes associated with stress response.

This research found, in contrast to WT and UL, the chlorophyll content of *FaTINY2* transgenic *A. thaliana* was steadier after treatment with low temperature and salt, and there was a minor rise in MDA, O_2_^−^ and H_2_O_2_ levels, alongside a slight increase in proline and the activities of CAT, SOD, and POD saw a substantial rise. The findings indicate that exposure to cold and salt stress may trigger *FaTINY2* to engage in the abiotic stress response mechanism of plants. In addition, the overexpression of *FaTINY2* could enhance plant stress resistance through multiple pathways, including influencing plant antioxidant mechanisms and altering the expression of subsequent stress-associated genes through the regulation of the CBF and ABA pathways, thereby enhancing tolerance of transgenic plants to both low-temperature and high salt stress (Appendix A). Given that *A. thaliana* is a model plant, the study focused on the varied roles of *FaTINY2* in *A. thaliana*. Whether *FaTINY2* can perform a similar function in strawberry remains to be further verified. Findings from this research indicated that *FaTINY2* was capable of effectively influencing *A. thaliana* resistance to cold and salt conditions, which provided a potential theoretical basis for strawberry cultivation in cold and saline-alkali regions.

## 4. Materials and Methods

### 4.1. Plants Materials

Octoploid cultivated strawberry (*Fragaria* × *ananassa*) was used as experimental material in this study. Materials selected from Northeast Agricultural University, Harbin, China. The seeds were seeded in a mixed substrate with a ratio of 2:1:1 nutrient soil, vermiculite, and perlite and cultured in an incubator at a constant temperature of 22 °C, with light for 16 h and darkness for 8 h (relative humidity of 70%) [73].

### 4.2. Plant Growth Situation and Treatment

Thirty seedlings with thriving were selected and divided into 6 groups, with 5 seedlings in each group, and subjected to stress treatment [73,74]. The first group, as the control group, was placed in a constant temperature incubator (22 °C). The other 5 groups were treated with 4 °C, 200 mM NaCl, 15% PEG6000, 37 °C, and 100 µM ABA successively to simulate low temperature, high salt, drought, high temperature, and ABA stress environment, respectively [75]. Following stress treatments of 0, 1, 2, 4, 6, 8, and 12 h, the leaves and roots of the seedlings were selected and rapidly submerged in liquid nitrogen and preserved at −80 °C to ensure the normal extraction of RNA.

### 4.3. RNA Extraction and Cloning of FaTINY2

Using young leaves and roots of strawberry seedlings as experimental materials, total RNA was isolated using an OminiPlant RNA Kit (Conway Collection, Beijing, China). Subsequently, the Universal Plant Total RNA Isolation Kit from Vazyme (Nanjing, China) was utilized for purification. The Takara first-strand cDNA Synthesis SuperMixture kits, employing RNA as a template, facilitated the reverse transcription of the first-strand cDNA. Furthermore, a pair of specific primers (*FaTINY2*-F/R) were designed. By employing cDNA as a template, PCR was used to amplify the target regions, followed by linking the PCR product to the ASY-T1 vector (TransGen Biotech, Beijing, China), and the sequence was obtained after positive detection.

### 4.4. Subcellular Localization Analysis of FaTINY2

A pair of primers with BamH I and Xba I enzyme digestion sites (FaTINY2-slF/slR; Appendix A) were used to amplify the FaTINY2 fragment with enzyme digestion sites by PCR amplification. The transient expression vector of FaTINY2-GFP was constructed by double digestion of PCR products and pCAMBIA1300 vector with BamH I and Xba I restriction endonucleases. Subsequently, the constructed transient expression vector pCAMBIA1300-FaTINY2-GFP was transformed into *Agrobacterium* strain GV3101, and 35s::FaTINY2-GFP and 35s::GFP were transformed using the same method. Then, the transformed *Agrobacterium* solution was injected into the outer epidermal cells of the tobacco leaf. The plants underwent cultivation at 24 °C over a period of 3 d, followed by the observation of their fluorescence signals through confocal microscopy (Zeiss, Oberkochen, Germany).

### 4.5. Sequence Analysis and Structure Prediction of FaTINY2

Through ExPASy-ProtParam (https://web.expasy.org/protparam/, accessed on 6 January 2023), obtained the primary structure of FaTINY2. NCBI (https://www.ncbi.nlm.nih.gov/, accessed on 6 January 2023) collected the amino acid sequences of FaTINY2 protein and the TINY proteins of other plants. These sequences were compared using DNAMAN 9.0 software, and MEGA7.0 was used to construct a phylogenetic tree. SMART (http://smart.embl-heidelberg.de/, accessed on 6 January 2023), SOPMA (https://npsa.lyon.inserm.fr/cgi-bin/npsa_automat.pl?page=/NPSA/npsa_sopma.html, accessed on 6 January 2023) and SWISS-MODEL (https://swissmodel.expasy.org/, accessed on 13 January 2023) were used to predict the domain, secondary structure and tertiary structure of FaTINY2, respectively.

### 4.6. Expression Analysis of FaTINY2

The qPCR primers (*FaTINY2*-qF/qR; Appendix A) were designed. The expression level of *FaTINY2* in different tissues was detected by qPCR. Furthermore, the construction of the qPCR reaction system was based on the methodology by Li [75]. In this system, the corresponding primers were designed (*FaActin*-F/R; Appendix A), and *FaActin* was the internal reference gene. The 2^−∆∆Ct^ method was used to estimate the expression level of target genes [76].

### 4.7. Construction of Transgenic Arabidopsis thaliana

*FaTINY2*-F and *FaTINY2*-R primers (*FaTINY2*-F/R; Appendix A) were used to amplify *FaTINY2* cDNA and connect *FaTINY2* cDNA to BamH I and Xba I of pCAMBIA1300 vector to construct *FaTINY2*-OE vector. The *FaTINY2*-OE vector was transferred into *Agrobacterium* GV3101 and an inflorescence-mediated method was used to transform *A. thaliana* (Col-0) [77]. For the examination of transgenic strains, MS medium supplemented with 50 mg/L kanamycin was utilized [78,79]. Wild type (WT) and unloaded line (UL) were used to identify transgenic lines by qRT-PCR. The T3 transgenic line was utilized in the subsequent analysis [80].

### 4.8. Stress Treatment of Arabidopsis thaliana

WT, UL, and transgenic lines (L1, L4, L5) were used as experimental materials for culture. After the cotyledon was exposed, the *A. thaliana* seedlings were transplanted into pots containing a mixed substrate (soil:vermiculite:perlite = 2:1:1) [81]. Seedlings exhibiting the same growth conditions were chosen and split into two groups, each containing 20 seedlings and 4 in a single pot. The first group of seedlings was incubated at −8 °C for 14 h and then transferred to normal room temperature (24 °C) for 7 days. The second group of seedlings was treated with high salt for 7 days and irrigated with 200 mM NaCl solution every 2 days. Then irrigated with fresh water for 3 days.

### 4.9. Determination of Related Physiological Indexes of Arabidopsis thaliana

Measurements were taken of the survival rates and physiological indexes for the *A. thaliana* lines (WT, UL, L1, L4, L5). Spectrophotometry was employed to quantify chlorophyll contents [82]. Zhang method was used to measure the activities of antioxidant enzymes such as CAT, SOD, and POD [83]. The contents of H_2_O_2_ and O_2_^−^ were ascertained using diaminobenzidine (DAB) and nitroblue tetrazole (NBT), respectively [84]. The content of malondialdehyde in the specimen was ascertained using the TBA method [85]. The content of proline was determined by the sulfosalicylic acid method [86].

### 4.10. Expression Analysis of Cold and Salt Stress-Related Response Genes

Using *AtActin* as an internal reference gene, the qPCR primers (*AtActin*-qF/qR; Appendix A) were designed. The qPCR method was performed on WT, UL, L1, L4, and L5 to determine the expression levels of abiotic stress-related response genes [87]. The expression of target genes was estimated by 2^−∆∆Ct^ [76].

### 4.11. Statistical Analysis

All of the data we studied was collected from three technical replicates. The mean values of the repeated trials were used as the values of the corresponding sample. SPSS 21.0 was used for significance analysis, and one-way ANOVA was used for Student’s *t*-test. Significant statistical differences were denoted by the notations * *p* ≤ 0.05, ** *p* ≤ 0.01.

## 5. Conclusions

In this study, nuclear localization of DREB TF gene *FaTINY2* was successfully cloned from octaploid strawberry *Fragaria* × *ananassa*, and explored its expression level and regulatory mechanism under normal and abiotic stress conditions. The study found that *FaTINY2* showed significant sensitivity to cold and salt stress, and quantitative analysis confirmed that the gene was tissue-specific. Subsequent studies indicated that *FaTINY2* could influence plant stress reactions by regulating antioxidant enzyme activities and the activation of genes linked to resistance downstream. The findings imply that *FaTINY2* could be crucial in reg regulating plant tolerance to abiotic stress.

## Figures and Tables

**Figure 1 ijms-26-02109-f001:**
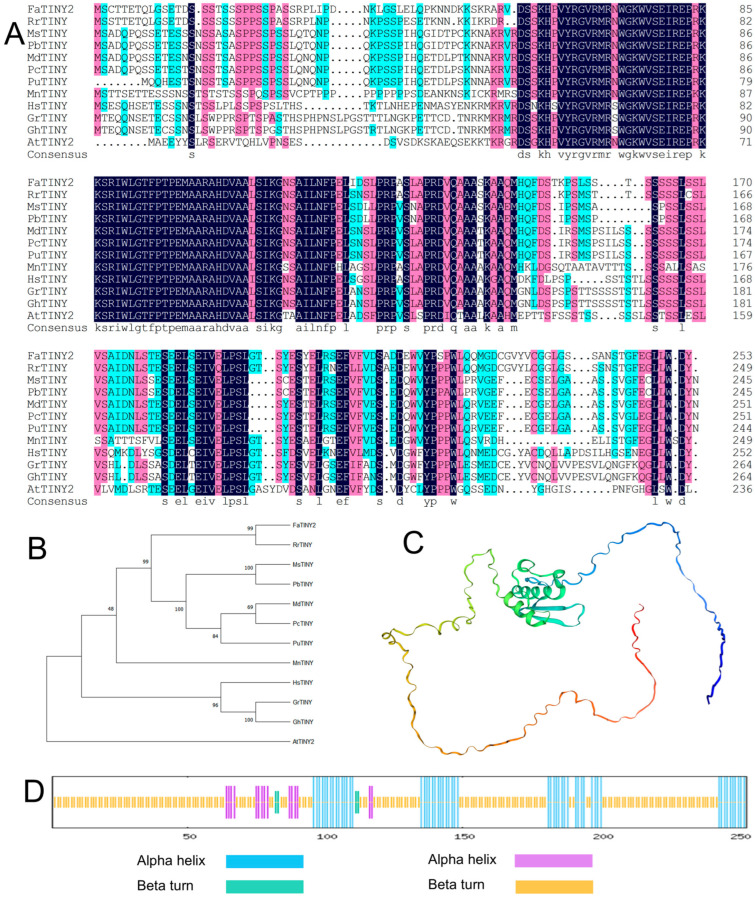
Analyzing the differences and evolutionary links between FaTINY2 and TINY transcription factors of other various species, along with forecasting the domains and structure of FaTINY2 proteins. (**A**) Comparative analysis of amino acid sequences of FaTINY2 protein and other plant TINY proteins (**B**) Phylogenetic tree analysis of TINY protein in *Fragaria* × *ananassa* and other plants. The accession numbers are as follows: FaTINY2 (*Fragaria* × *ananassa*, AZL19427.1), RrTINY (*Rosa rugosa*, XP_062025076.1), MsTINY (*Malus sylvestris*, XP_050123751.1), PbTINY (*Pyrus* × *bretschneideri*, XP_009367456.2), MdTINY (*Malus domestica*, NP_001292812.1), PcTINY (*Pyrus communis*, XP_068314191.1), PuTINY (*Pyrus ussuriensis* × *Pyrus communis*, KAB2629625.1), MnTINY (*Morus notabilis*, XP_010102390.1), HsTINY (*Hibiscus syriacus*, XP_039017109.1), GrTINY (*Gossypium raimondii*, XP_012447312.1), GhTINY (*Gossypium hirsutum*, XP_040949117.1), AtTINY2 (*Arabidopsis thaliana*, OAO95125.1) (**C**) Tertiary structure of FaTINY2 protein predicted by SWISS-MODEL. (**D**) Prediction of protein secondary structure by SOPMA.

**Figure 2 ijms-26-02109-f002:**
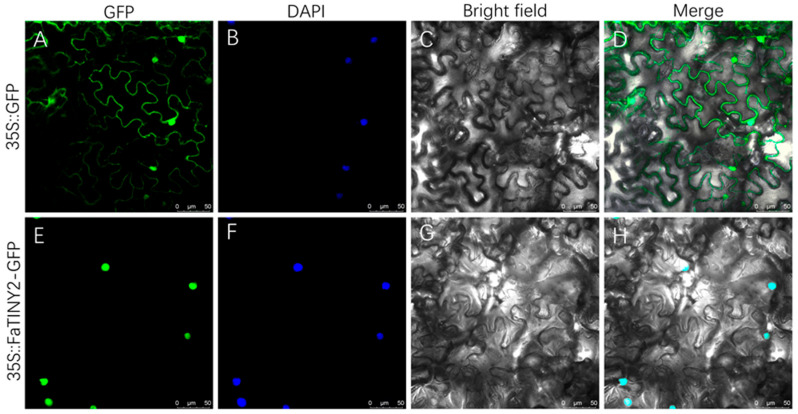
Subcellular localization of FaTINY2 in epidermal cells of tobacco leaves. Plasmid 35S::GFP and 35S::FaTINY2-GFP were injected into the cells by *Agrobacterium* injection method. (**A**,**E**) GFP fluorescence. (**B**,**F**) DAPI staining. (**C**,**G**) Bright-field images. (**D**,**H**) Merged. Bar = 50 µm.

**Figure 3 ijms-26-02109-f003:**
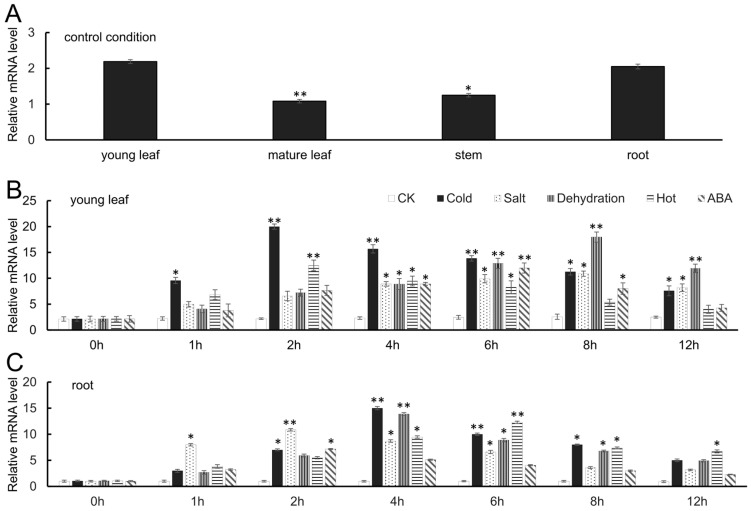
Expression pattern analysis of *FaTINY2* in *Fragaria* × *ananassa*. (**A**) Levels of *FaTINY2* expression across various tissues in control conditions (**B**) Levels of *FaTINY2* expression in young leaves, under control conditions and under five varying stress conditions, at distinct intervals (**C**) Levels of *FaTINY2* expression in roots, under control conditions and under five varying stress conditions, at distinct intervals. The error bars represent the standard deviation. Data represent the means ± SD of triplicate experiments. The presence of asterisks over the error bars signifies a notable disparity between the experimental and control groups (Student’s *t*-test; * *p* ≤ 0.05, ** *p* ≤ 0.01).

**Figure 4 ijms-26-02109-f004:**
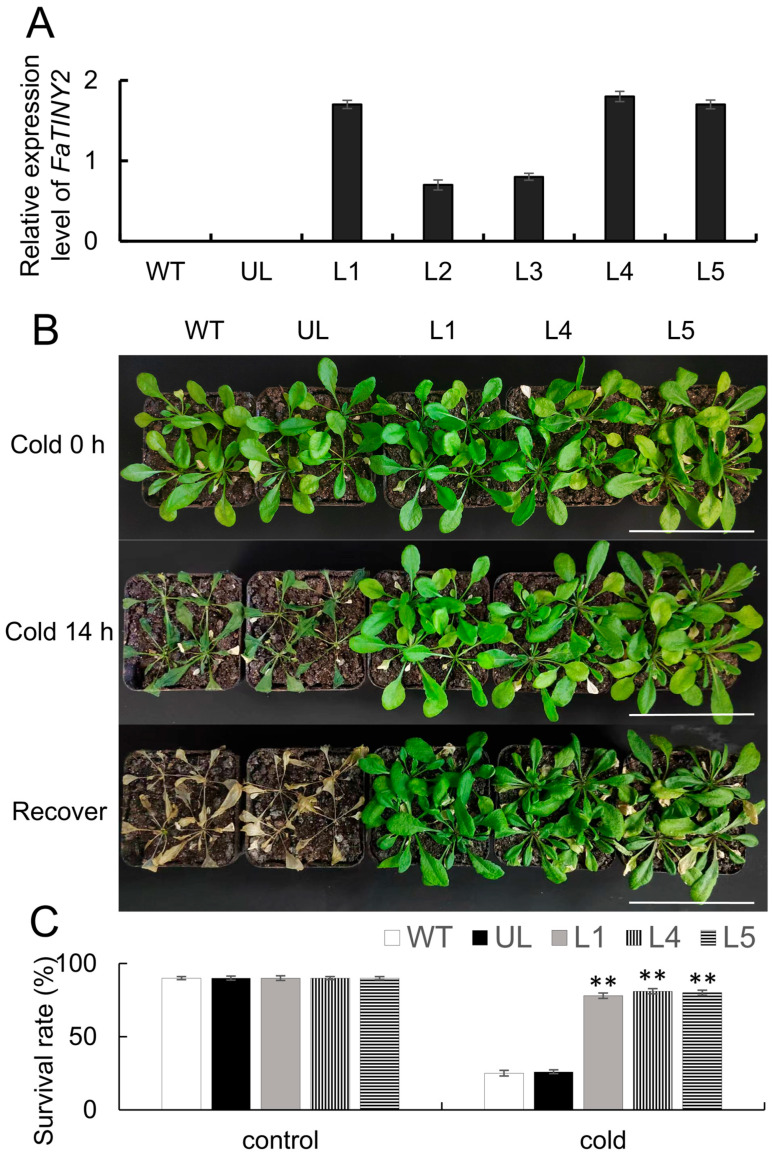
Cultivation of *FaTINY2* transgenic *A. thaliana* in low-temperature environments. (**A**) Comparative levels of *FaTINY2* expression in WT, UL, and five lines overexpressing *FaTINY2* (L1, L2, L3, L4, L5). (**B**) Phenotype variations in wild type (WT) and unloaded line (UL) and *FaTINY2*-overexpressing lines (L1, L2, L3, L4, L5) under the control environment (22 °C), cold treatment (−8 °C for 14 h), and after recovery room temperature. Bar = 5 cm. (**C**) Survival rates for WT, UL, and transgenic lines in both the control environment and under cold conditions. Data represent the means ± SD of triplicate experiments. Indicator levels in WT were used as controls. The presence of asterisks above each error bar signifies a significant difference among the various lines (One-way ANOVA, ** *p* ≤ 0.01).

**Figure 5 ijms-26-02109-f005:**
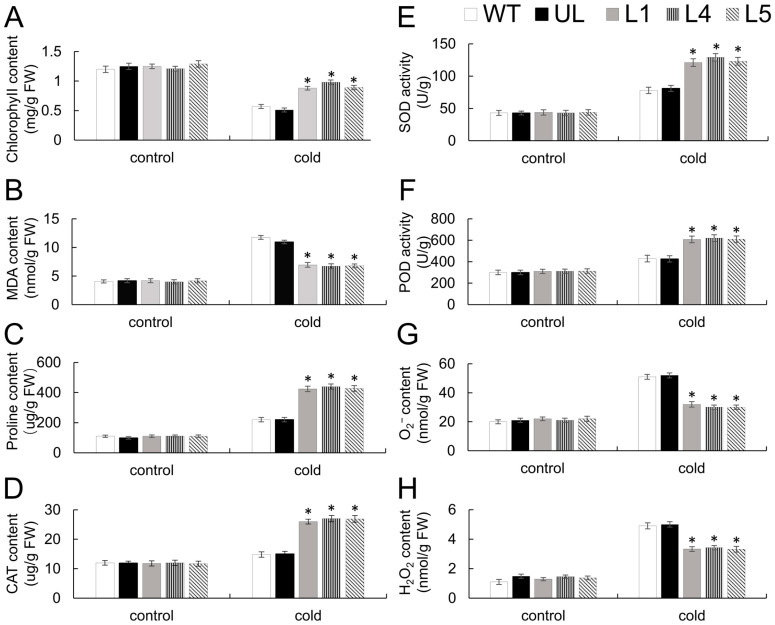
Indicators of physiology in *FaTINY2* transgenic *A. thaliana* plants subjected to low-temperature environments. The contents of (**A**) Chlorophyll, (**B**) MDA, (**C**) Proline, (**G**) O_2_^−^ and (**H**) H_2_O_2_, and the activities of (**D**) CAT, (**E**) SOD and (**F**) POD were observed under non-stress conditions (22 °C) and cold treatment (−8 °C, 14 h). Data represent the means ± SD of triplicate experiments. Indicator levels in WT were used as controls. The presence of asterisks above each error bar signifies a significant difference among the various lines (One-way ANOVA, * *p* ≤ 0.05).

**Figure 6 ijms-26-02109-f006:**
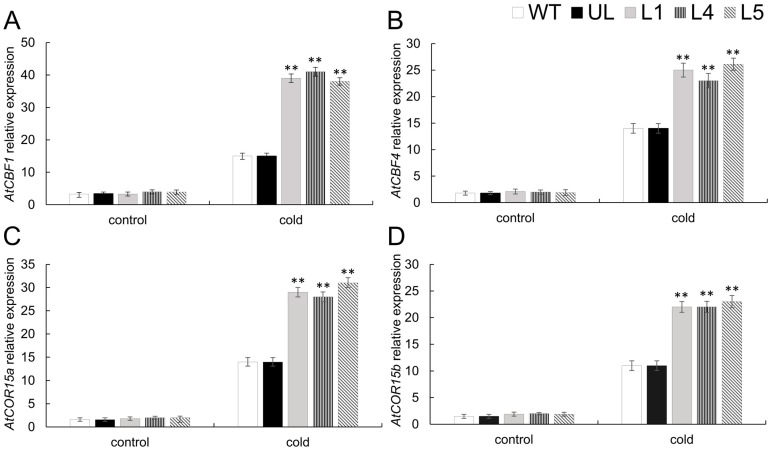
Gene expression levels linked to cold stress in transgenic A. thaliana overexpressing *FaTINY2* during exposure to low-temperature environments. Levels of expression for (**A**) *AtCBF1*, (**B**) *AtCBF4*, (**C**) *AtCOR15a*, and (**D**) *AtCOR15b* across WT, UL, and transgenic lines (L1, L4, L5). Data represent the means ± SD of triplicate experiments. Indicator levels in WT were used as controls. The presence of asterisks above each error bar signifies a significant difference among the various lines (One-way ANOVA, ** *p* ≤ 0.01).

**Figure 7 ijms-26-02109-f007:**
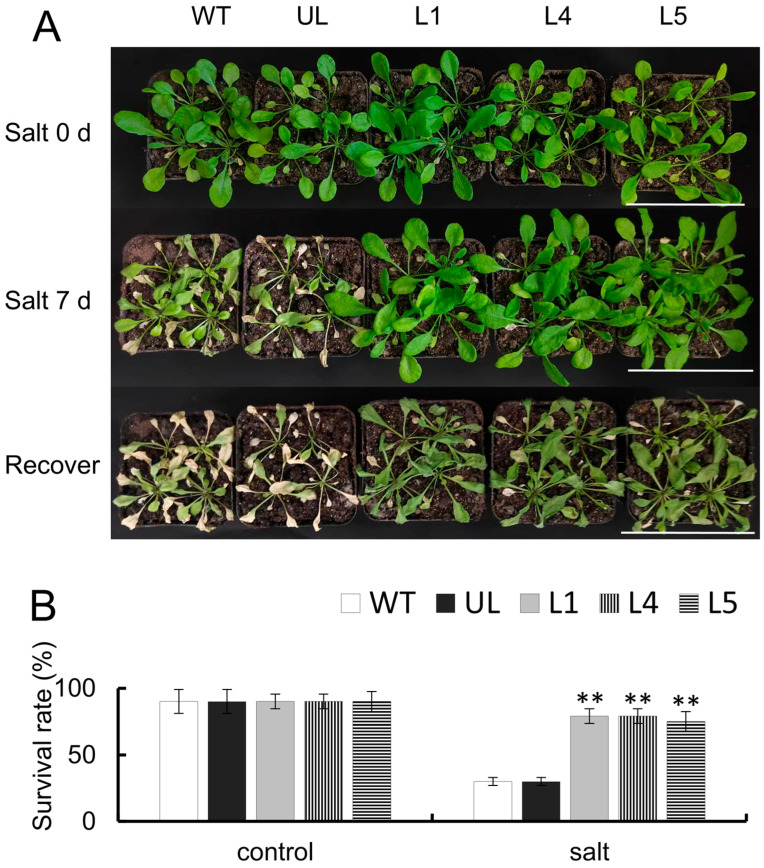
Cultivation of *FaTINY2* transgenic *A. thaliana* in high salt environments. (**A**) Phenotype variations in wild type (WT) and unloaded line (UL), and *FaTINY2*-overexpressing lines (L1, L2, L3, L4, L5) under the control environment, salt treatment (200 mM NaCl irrigation for 7 days, a total of 4 times), and restored water irrigation. Bar = 5 cm. (**B**) Survival rates for WT, UL, and transgenic lines in both the control environment and under salt treatment. Data represent the means ± SD of triplicate experiments. Indicator levels in WT were used as controls. The presence of asterisks above each error bar signifies a significant difference among the various lines (One-way ANOVA, ** *p* ≤ 0.01).

**Figure 8 ijms-26-02109-f008:**
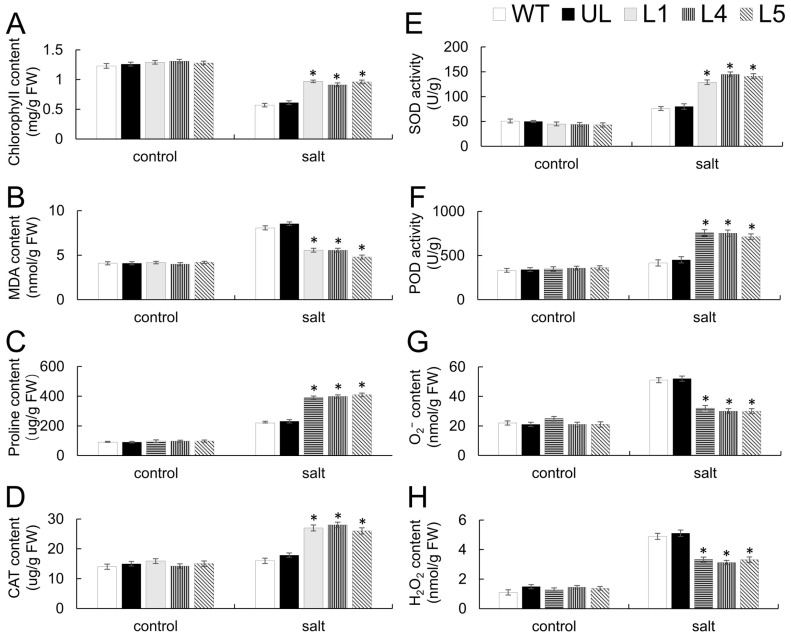
Indicators of physiology in *FaTINY2* transgenic *A. thaliana* plants subjected to high-salt treatment. The contents of (**A**) Chlorophyll, (**B**) MDA, (**C**) Proline, (**G**) O_2_^−^ and (**H**) H_2_O_2_, and the activities of (**D**) CAT, (**E**) SOD and (**F**) POD were observed under non-stress conditions and salt treatment (200 mM NaCl for 7 d). Data represent the means ± SD of triplicate experiments. Indicator levels in WT were used as controls. The presence of asterisks above each error bar signifies a significant difference among the various lines (One-way ANOVA, * *p* ≤ 0.05).

**Figure 9 ijms-26-02109-f009:**
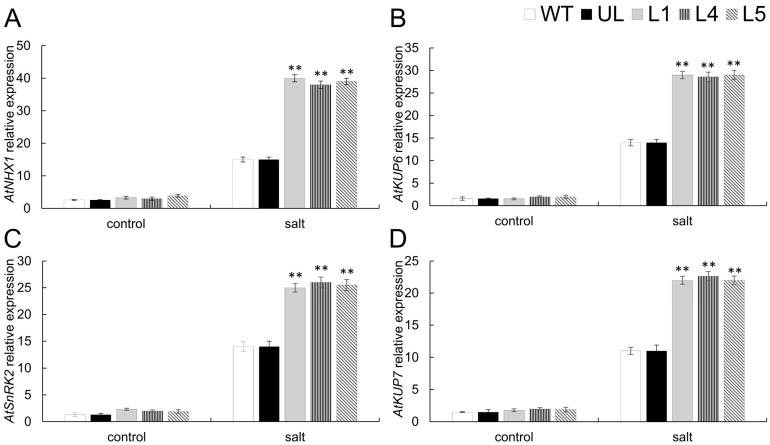
Gene expression levels linked to cold stress in transgenic *A. thaliana* overexpressing *FaTINY2* expression under high salt conditions. Levels of expression for (**A**) *AtNHX1*, (**B**) *AtKUP6*, (**C**) *AtSnRK2*, and (**D**) *AtKUP7* across WT, UL, and transgenic lines (L1, L4, L5). Data represent the means ± SD of triplicate experiments. Indicator levels in WT were used as controls. The presence of asterisks above each error bar signifies a significant difference among the various lines (One-way ANOVA, ** *p* ≤ 0.01).

## Data Availability

Data are contained within the article and Appendix A.

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
