# Peer review of "Overexpression of a Fragaria × ananassa AP2/ERF Transcription Factor Gene (FaTINY2) Increases Cold and Salt Tolerance in Arabidopsis thaliana"

_ijms, 2025, doi:10.3390/ijms26052109_

Round 1
Reviewer 1 Report
Comments and Suggestions for Authors
Results
Fig 1- ....(B) Analysis phylogenetic tree in ...(line121). It is not clear what the authors mean. Please rephrase.
Expression Analysis of FaTINY2
- The explanation text should be revised and simplified. The sentences from 137 to 141 can be replaced with something similar to : the further DAPI staining confirmed the nuclear localization of the FaTINY2 protein.
- The scale bar is not visible in figure 2
Expression Analysis of FaTINY2
- The use of the expression "concentration" should be replaced by "expression level"
- There are several mistakes in the text, namely when referring to the time points.
- What do the authors mean by:" indicators levels in WT were used as controls."? (line 167)
Overexpression of FaTINY2 Enhanced Cold Tolerance of Arabidopsis Thaliana
- The explanation text should be revised and simplified.
- Why did the authors change the temperature for the cold treatment from 4o to -8ºC?
- When did the material for biochemical analysis was collected? Immediately after exposure to cold or after the recovery period?
- Given that one of the effects of cold and salt stress is the loss of water by the tissues, the results shown in Figures 5 and 8 should be expressed as /g DW.
- At what temperature were the activity tests carried out?
Discussion
"Findings indicated a significantly increase in FaTINY2 expression..." (line 284) should be replaced by "Findings indicated that FaTINY2 is more expressed in....".
The authors can not related this expression levels with sensitivity to stress. This is just a tissued specific expression.
Comments on the Quality of English Language
The language used in the article needs to be revised to make it easier to understand the results presented.
The language used in the article needs to be fully revised to make it easier to understand the results presented.
Expressions such as :“...expression in the concentration” (line 149), "...signifies a notable disparity...." (line 168) or " PCR was used to magnify..." (line 381), are not correct from a scientific point of view.
Author Response
Dear Editor Nichola Whittingham and Reviewers:
Thanks to Reviewers for their time and effort in reviewing the manuscript. We are very grateful to Reviewer for reviewing the paper so carefully.
Responds to the reviewers' comments:
Reviewer #1:
About Results:
Comment: Fig 1 ....(B) Analysis phylogenetic tree in ...(line121). It is not clear what the authors mean. Please rephrase.
Response: We appreciate this good suggestion and we have revised this sentence to ‘Phylogenetic tree analysis of TINY protein in Fragaria × ananassa and other plants.’
Expression Analysis of FaTINY2
Comment: The explanation text should be revised and simplified. The sentences from 137 to 141 can be replaced with something similar to : the further DAPI staining confirmed the nuclear localization of the FaTINY2 protein.
Response: We appreciate this good suggestion and we have revised this sentence.
Comment: The scale bar is not visible in figure 2
Response: Thank you for your this question, the scale is at the bottom right of each figures.
Expression Analysis of FaTINY2
Comment: The use of the expression "concentration" should be replaced by "expression level"
Response: Thanks for your suggestion, we have changed the 'concentration' to 'expression level'.
Comment: There are several mistakes in the text, namely when referring to the time points.
Response: Thanks for your suggestions, we have carefully checked and corrected the errors in the time points, including the text and pictures.
Comment: What do the authors mean by:" indicators levels in WT were used as controls."? (line 167)
Response: We are very sorry for our incorrect writing and this sentence should be deleted.
Overexpression of FaTINY2 Enhanced Cold Tolerance of Arabidopsis Thaliana
Comment: The explanation text should be revised and simplified.
Response: Thanks for your suggestions, we have simplified and modified this part.
Comment: Why did the authors change the temperature for the cold treatment from 4ºC to -8ºC?
Response: The expression level of FaTINY2 in strawberry tissues was investigated under the cold treatment condition of 4℃. When the expression level of FaTINY2 in transgenic Arabidopsis thaliana was investigated, the cold treatment temperature was -8℃. This is because at -8℃, the expression level of this gene in Arabidopsis Thaliana is more significant.
Comment: When did the material for biochemical analysis was collected? Immediately after exposure to cold or after the recovery period?
Response: The experimental materials were collected immediately after exposure to cold.
Comment: Given that one of the effects of cold and salt stress is the loss of water by the tissues, the results shown in Figures 5 and 8 should be expressed as /g DW.
Response: Thank you for your suggestion, but we also consulted some literature and found that FW was used as experimental material.
Comment: At what temperature were the activity tests carried out?
Response: The activity of physiological indexes was measured at room temperature.
Discussion
Comment: "Findings indicated a significantly increase in FaTINY2 expression..." (line 284) should be replaced by "Findings indicated that FaTINY2 is more expressed in....".
Response: We strongly agree with your suggestion and we reviesed this sentence.
Comment: The authors can not related this expression levels with sensitivity to stress. This is just a tissued specific expression.
Response: Under the same control conditions, young leaves and roots had the highest expression levels compared to older leaves and stems indicating that they are more sensitive to external responses.

Reviewer 2 Report
Comments and Suggestions for Authors
Dear authors,
Li et al documents that overexpression of a Fragaria × ananassa AP2/ERF transcription factor gene increases cold and salt tolerance in Arabidopsis thaliana. The authors cloned the gene from Fragaria × ananassa and identified its bioinformatic characteristics. They found that this gene is localized in the nucleus and tissue specific in Fragaria × ananassa. To confirm its function further, they transformed this gene into Arabidopsis, observed the altered phenotypes under abiotic stresses. Moreover, they analyzed the contents of ROS, MDA, SOD, CAT, etc. and the expression levels of the gene under abiotic stresses. Then the authors analyzed the expression patterns of the genes involved in the cold pathway. Based on the above results, they suggested that FaTIN2 plays a role in abiotic stresses.
The following comments are suggestions that are important to address to improve the overall quality of this manuscript.
Major comments:
- Check expression level in Fig.4 A, calculate expression levels of Ls compared to the level of WT.
Minor comments:
- Line 27. Tolerant ‘to’ instead of ‘of’.
- Line 105. Add ‘comma’ before Da.
- Line 108. It is classified as ……
- Line 132. In order to ……., we construct …….
- Line 149. Add ‘comma’ in front of ‘Findings’. Two many ‘Findings indicated’, please use the other way to present results.
- Line 151. ‘LT’ ?
- Line 176. 14 h?
- Line 167. Indicators to ‘indicator’.
- 4. Panel A.
- 4 to 9. One-way ANOVA instead of student’s t-text.
- 7 200 mM NaCl irrigation for 7 days. How many times irrigation is in 7 days? Please clarify in the legend.
- 8 delete ‘Chlorophyll’ ,…… as they are in the y axis, respectively.
- 10 had better be as a supplementary fig.
Author Response
Dear Editor Nichola Whittingham and Reviewers:
Thanks to Reviewers for their time and effort in reviewing the manuscript. We are very grateful to Reviewer for reviewing the paper so carefully.
Responds to the reviewers' comments:
Reviewer #2:
About Major comments:
Comment: Check expression level in Fig.4 A, calculate expression levels of Ls compared to the level of WT.
Response: There is no FaTINY2 gene in wild Arabidopsis thaliana, so the expression of this gene in WT is 0.
About Minor comments:
Comment 1: Line 27. Tolerant ‘to’ instead of ‘of’.
Response: Thanks for your suggestion, we have changed the 'of' to 'to' in Line 27
Comment 2: Line 105. Add ‘comma’ before Da.
Response: Thanks for your suggestion, we have add ‘comma’ before Da.
Comment 3: Line 108. It is classified as ……
Response: we have changed the 'classifies' to 'classified' in Line 108
Comment 4: Line 132. In order to ……., we construct …….
Response: Thank you for your suggestion, we have revised this sentence, these changes will not affect the meaning of the sentence.
Comment 5: Line 149. Add ‘comma’ in front of ‘Findings’. Two many ‘Findings indicated’, please use the other way to present results.
Response: Thank you for your suggestion, we have added a ‘comma’ in front of ‘Findings’.
Commen 6t: Line 151. ‘LT’ ?
Response: LT means ‘low temperature’, the abbreviation first appears in line 35 of this article.
Comment 7: Line 176. 14 h?
Response: Yes, cold treatment is 14h, and 7d in the figure is a writing error, which we have modified.
Comment 8: Line 167. Indicators to ‘indicator’.
Response: Thank you for your suggestion, The description here is not standard, so this sentence should be deleted.
Comment 9: 4. Panel A.
Response: Thank you for your review of the figure. We have not fully understood the problem you pointed out. Could you please explain it in detail so that we can conduct in-depth analysis and adjustment based on it.
Comment 10: 4 to 9. One-way ANOVA instead of student’s t-text.
Response: Thank you for your suggestion, and we have changed the 'student’s t-text' to 'One-way ANOVA''
Comment 11: 7 200 mM NaCl irrigation for 7 days. How many times irrigation is in 7 days? Please clarify in the legend.
Response: Under salt treatment conditions, the seedlings were treated with high salt in 7 days and irrigated with 200 mM NaCl solution every 2 days. Therefore, NaCl solution was used for irrigation on days 1, 3, 5, and 7, respectively, for a total of four times.
评论 12:8 删除 '叶绿素' ,......因为它们分别位于 y 轴上。
回应:我们很感激这个好建议,我们已经更新了图 5 和图 8。
评论 13:10 最好作为补充图。
回应:感谢您的建议,我们已将图 10 用作补充图。
